# Asymptomatic Bacteriuria (ABU) in Elderly: Prevalence, Virulence, Phylogeny, Antibiotic Resistance and Complement C3 in Urine

**DOI:** 10.3390/microorganisms9020390

**Published:** 2021-02-14

**Authors:** Rikke Fleron Leihof, Karen Leth Nielsen, Niels Frimodt-Møller

**Affiliations:** 1Department of Microbiology and Infection Control, Statens Serum Institut, 2200 Copenhagen, Denmark; rikke@leihof.dk; 2Department of Clinical Microbiology, Rigshospitalet, 2100 Copenhagen, Denmark; karen.leth.nielsen.01@regionh.dk

**Keywords:** asymptomatic bacteriuria, urinary tract infection, *Escherichia coli*, virulence, phenotype, complement C3, whole-genome sequencing

## Abstract

**Background**: The incidence of asymptomatic bacteriuria (ABU) increases with age and is most common for persons 80 years of age and above and in elderly living in nursing homes. The distinction between ABU and urinary tract infection (UTI) is often difficult, especially in individuals, who are unable to communicate their symptoms, and there is a lack of objective methods to distinguish between the two entities. This can lead to overuse of antibiotics, which results in the selection and dissemination of antibiotic resistant isolates. **Materials and methods:** From voided midstream urine samples of 211 participants ≥60 years old from nursing homes, an activity center and a general practitioners clinic, we collected 19 ABU, 16 UTI and 22 control urine samples and compared them with respect to levels of complement component C3 in urine as determined by an ELISA assay relative to creatinine levels in the same urine samples, as measured by a creatinine assay. Further, we studied all *Escherichia coli* isolates for selected virulence genes by multiplex PCR, and by whole-genome sequencing (WGS) for genotypes and phylogenetic clustering. Antibiotic susceptibility was determined by microtiter broth dilution. **Results:** We identified a prevalence of ABU of 18.9% in nursing home residents, whereas ABU was only found in 4% of elderly living in the community (*p* < 0.001). *E. coli* from ABU patients were significantly more antibiotic resistant than *E. coli* from UTIs (*p* = 0.01). Prevalence of classical virulence genes, detected by multiplex PCR, was similar in *E. coli* isolates from ABU and UTI patients. Whole-genome sequencing of the *E. coli* isolates showed no specific clustering of ABU isolates compared to UTI isolates. Three isolates from three different individuals from one of the nursing homes showed signs of transmission. We demonstrated a significantly increased level of C3/creatinine ratio in ABU and UTI samples compared to healthy controls; however, there was no significant difference between the ABU and UTI group with respect to C3 level, or virulence factor genes. **Conclusion:** ABU was significantly more prevalent in the elderly residing in nursing homes than in the elderly living at home. Antibiotic resistance was more prevalent in *E. coli* from nursing homes than in UTI isolates, but there was no difference in prevalence of virulence associated genes between the two groups and no phylogenetic clustering, as determined by WGS relative to the two types of *E. coli* bacteriuria. The similar complement C3 response in ABU and UTI patients may indicate that ABU should be reconsidered as an infection albeit without symptoms.

## 1. Introduction

Asymptomatic bacteriuria (ABU) is particularly common in long-term care facilities (LTCFs) with a reported prevalence of 25–50% in the residents [1,2]. The frequency of ABU varies greatly with respect to sex, age, somatic condition and type of institution [2]. Several studies have addressed the appropriate management of asymptomatic bacteriuria in LTCF residents, repeatedly showing treatment to be unnecessary, since it does not decrease the prevalence of bacteriuria, or the frequency of symptomatic UTI episodes or incontinence [3,4,5,6]. In addition, treatment of ABU can lead to unwanted side effects, including increased antimicrobial resistance in recurrent infection, and should only be initiated in pregnant women or in cases where ABU can have adverse outcomes that could be prevented by antimicrobial treatment [5,7].

Determining whether a LTCF resident has a symptomatic UTI or ABU can be challenging, since LTFC residents often have hearing impairments, dementia, confusion, or other underlying conditions that can interfere with the assessment of new or altered symptoms [1]. Some of the most common reasons for suspecting UTI in nursing home residents therefore include changes in mental state (lethargy, disorientation, restlessness, increased irritability and aggressiveness), or the new onset of confusion and delirium [8,9,10]. These nonspecific symptoms can, however, also be related to other conditions and are not predictive for UTI even in the presence of a positive urine sample [11]. The diagnostic uncertainties regarding UTI in the elderly population commonly lead to overuse of antimicrobials with the risk of increasing antimicrobial resistance [1].

*Escherichia coli* is the dominant pathogen in both UTI and ABU, and several investigators have tried to determine whether there are differences in the virulence potential in *E. coli* between the two entities, which could help differentiate ABU from UTI [12,13,14,15]. In one study the typically prevalent phylotypes B2 and D in UTI were significantly less common among ABU patients [12], but this was not reproduced in other studies, where various molecular biological methods were employed to investigate phylotyping and genotyping in similar patient groups [13,14,15].

The aim of the present study was to investigate the prevalence of ABU in the elderly, both in subjects attending an activity center but residing at home, and in nursing home residents. Further, we investigated the *E. coli* isolates from ABU cases compared with isolates from UTI cases for antibiotic resistance; the presence of typical virulence genes by PCR; and phylogroups and phylogenetic relationships based on SNP distances by whole-genome sequencing. Lastly, we measured complement C3 concentrations in urine from patients with ABU and UTI in order to evaluate a possible correlation between urinary levels of complement C3 and ABU. If urinary concentration of complement C3 is predictive of UTI or ABU in the elderly, this could improve the discrimination between these two conditions with a reduction in unnecessary use of antibiotics as a result.

## 2. Materials and Methods

### 2.1. Study Participants and Sampling

The study was approved by the Human Research Ethics Committee of the Capital Region of Copenhagen (H-1-2013-078, 13 September 2013). Written informed consent was obtained from all participants.

Study participants were recruited prospectively from two nursing homes (N = 49), an activity center for elderly (N = 140) and a general practice (GP) in Zealand, Denmark in 2013–2014 (N = 22). The population of study participants included elderly men and women ≥60 years with or without symptoms of UTI. Excluded were: patients with a urinary catheter, patients in renal dialysis, patients with progressed dementia or diagnosed cancer in the urinary system and patients undergoing antibiotic treatment at the time of the urine sampling or in a period of three months prior to the urine sampling. All study participants from nursing homes and the activity center were asked to answer a brief questionnaire concerning their sex, age and symptoms. Symptoms included frequent urination, burning sensation during urination, pain in the bladder region and/or lower back and/or fever. Participants from nursing homes were asked for additional urine samples three months after the first samples, in order to determine ABU prevalence.

According to the IDSA guidelines [16], we defined ABU as two consecutively voided urine samples (min. 24 h apart) with ≥10^5^ CFU/mL of the same bacterial strain, when the participant did not experience any symptoms. UTI was defined as one urine specimen with ≥10^3^ CFU/mL of typical urinary pathogens, when the participant experienced symptoms.

All participants were asked to use intimate washing wipes before delivering a midstream urine sample, which was analyzed by dipstick and cultured on a Flexicult agar plate (Diagnostica, Hillerød, Denmark) for cultivation overnight, and the remaining sample was stored at 5 °C for a maximum of three days until being frozen (−80 °C). All study participants with ≥10^5^ CFU/mL in urine samples and no UTI symptoms were asked for an additional sample, collected within two weeks. If the second sample also showed growth ≥10^5^ CFU/mL with the same bacterial species, the case was considered as ABU. Participants with symptoms and ≥10^3^ CFU/mL were not asked for an additional sample. Participants without symptoms and bacterial growth as well as a negative dipstick analysis were used as controls.

### 2.2. Bacterial Isolates

Isolates from Flexicult plates were re-streaked on ID-flexicult agar plates (Diagnostica, Hillerød, Denmark) to confirm identity. In addition, isolate identity was confirmed by Matrix-assisted laser desorption/ionization-time of flight (MALDI-TOF, Bruker Nordic, Copenhagen, Denmark). Isolates were frozen and stored at −80 °C.

### 2.3. Antimicrobial Susceptibility Testing of E. coli

Minimum inhibitory concentrations (MIC) were determined for ampicillin, cefuroxime, ertapenem, gentamicin, ciprofloxacin, trimethoprim/sulfamethoxazole and tetracycline using a 96-well Sensititre plate (Trek GN3F, Thermo Fisher Scientific, Roskilde, Denmark) applying EUCAST breakpoints [17]. Isolates showing resistance towards cephalosporins or carbapenems were further tested for possible ESBL, AmpC or carbapenemase production using phenotypical antibiotic disk tests (Rosco Diagnostica, Albertslund, Denmark). *E. coli* ATCC 25922 was used as a quality control in all runs.

### 2.4. Virulence Genotyping of E. coli

*E. coli* isolates were screened for the presence of 28 virulence associated genes (VAGs) with known or suspected relevance to ExPEC pathogenesis, as described by Ejrnæs et al. [18] and Skjøt-Rasmussen et al. [19]. The VAGs represent six functional groups: (1) eight adhesins: *afa*/*draBC* (Dr binding adhesin), *bmaE* (blood group M fimbriae), *fimH* (type 1 fimbriae), *fogG* (F1C fimbriae), *gafD* (G fimbriae), *iha* (iron-regulated gene A homologue adhesion, siderophore receptor), *papAH* (P-fimbriae) and *sfa*/*focDE* (S fimbriae, F1C fimbriae); (2) four biofilm related VAGs: *agn43* (antigen 43), *agn43aCFT073* (antigen 43, allele a CFT073), *agn43bCFT073* (antigen 43 allele b CFT073) and *agn43K12* (antigen 43, allele K12); (3) five iron uptake VAGs: *chuA* (heme receptor), *fyuA* (yersiniabactin siderophore receptor), *iutA* (aerobactin siderophore receptor), *ireA* (iron-regulated element, siderophore receptor) and *iroN* (salmochelin siderophore receptor); (4) five protectins: *iss* (increased serum survival), *kpsM* II (group II capsule), *kpsM* II K2 (group 2 capsule, K2), *KpsM* III (group III capsule) and *traT* (serum resistance); (5) four toxins: *cdtB* (cytolethal distending toxin), *cnf1* (cytotoxic necrotizing factor 1), *hlyD* (alpha hemolysin) and *sat* (secreted autotransporter toxin); (6) two miscellaneous: *ibeA* (brain endothelium) and *usp* (uropathogenic specific protein). In particular, *fimH* and *iutA* are common virulence genes in *E. coli* found in the gut or in the urinary tract and are important for adherence and penetration intracellularly [20,21].

### 2.5. Whole-Genome Sequencing

The *E. coli* isolates (one phenotype per study participant) from the ABU (N = 11; the single *E. coli* from nursing home number 2 was not sequenced) and UTI (N = 12) groups, respectively, were whole-genome sequenced using Illumina technology (Miseq, Illumina Inc., San Diego, CA, USA) with 2 × 250 bp technology. DNA was purified using DNeasy Blood and Tissue kit (Qiagen, Hilden, Germany). The isolates were assembled using Velvetoptimizer version 2.2.6 [22] and annotated using Prokka version 1.12 [23].

Phylogroups were determined in silico based on Clermont et al. [24]. Phylogenetic relationship was analyzed by Parsnp version 1.2 and harvesttools version 1.3 [25] filtering for recombination. The isolates of the present study were also compared to UTI isolates collected in Zealand, Denmark as described in detail previously [26]. This collection was collected from otherwise healthy, pre-menopausal women, and hence, represent a younger population (av. age = 35.2 ± 7.3) in order to compare to the isolates of the present study where the age is average 82.1 ± 9.1 years. MLST types were identified with Seqsphere version 7 (Ridom GmbH, Germany).

### 2.6. Content of Complement C3 in Urine Samples

C3 was measured by an enzyme-linked immunosorbent assay (ELISA) specific for human C3 (Abnova, Taipei, Taiwan). All runs were performed according to kit instructions with a sample control. Standard curves with an R^2^-value < 0.98 were transformed to polynomials and C3 concentrations calculated from the quadratic equation. Levels of C3 were normalized to urinary creatinine using the Microvue creatinine assay (Quidel Corporation, San Diego, CA, USA, and hence, expressed as C3/Cr in a ratio of ng/mg.

### 2.7. Statistics

GraphPad Prism 8 (GraphPad software, San Diego, CA, USA) was used for all statistical analyses, with a *p*-value of <0.05 considered significant unless otherwise stated. Prevalence was reported as percentages with 95% confidence intervals (CI). Unpaired Student’s *t*-tests were used to evaluate the correlations of C3/Cr levels in the ABU, UTI and control groups (*p* < 0.017, Bonferroni corrected). Correlation between C3/Cr and leukocyturia was conducted by linear regression.

The number of antimicrobial resistances and virulence-associated genes were compared between the UTI and ABU groups using the two-tailed Mann–Whitney U test. Phylotypes were compared using a two-tailed Fisher’s exact test. *p* < 0.05 was considered significant.

## 3. Results

### 3.1. Prevalence of ABU

ABU was found in seven of the 49 participants in the first sample collection at nursing homes. Twenty of the participants provided an additional sample three months later, in order to determine whether the same patients had ABU at this time. Only five of the seven ABU participants provided new samples after three months, where two had ABU with the same bacterial species (*E. coli*), two were colonized by new species (one had *E. coli* the first time and *Klebsiella oxytoca* the second time; the other had *K. pneumoniae* the first time and *E. coli* the second time) and one had spontaneously cleared the colonization (*Proteus mirabilis*). In addition, three new participants with ABU were found during the second sample collection. Hence, a total of 14 (11 females, three males; 18.9%; (CI: 10.2%; 27.1%)) ABU samples were found from nursing home residents, i.e., a total of 74 residents sampled, 49 once and 25 twice (Figure 1). From the activity center, 140 individuals agreed to participate in the study. Of these, five had ABU (four females, one male) (4%; CI: 0.7%; 7.3%) (Figure 1). The difference in prevalence of ABU between the nursing homes residents and activity center individuals was significant (*p* < 0.0001). Sixteen patients with UTI were included in the study (15 females, one male; nursing homes N = 3, activity center N = 1 and 12 from the GP (10 GP patients had no growth in urine)) (Figure 1).

The median (range) of the ABU group was 79.5 years old (60–95 years old), and for the UTI group it was 84 years old (62–92 years), *p* > 0.05.

*E. coli* was the primary causative bacterium in both ABU and UTI groups, representing 12/19 cases (63%) and 12/16 cases (75%), respectively, of the two groups. Other isolates found included *K. oxytoca*, *K. pneumoniae*, *Enterococcus faecalis* and *P. mirabilis*.

### 3.2. Antimicrobial Resistance of E. coli Isolates

The prevalence of antibiotic resistance in the *E. coli* isolates is shown in Table 1. Two isolates from the ABU group were ESBL-producers. The ABU isolates were generally more resistant than the UTI isolates (24 resistance markers/12 isolates for ABU vs. 8/12 for UTI, *p* = 0.01 (Table 1)). However, an association between increased antimicrobial resistance and sample collection place was not found (data not shown).

### 3.3. Virulence Associated Genes in E. coli Isolates

The numbers of virulence associated genes identified in the ABU and UTI groups are visualized for genes identified in ≥4 of the isolates in one of the respective groups (Table 2). The virulence associated gene (VAG) *fimH* was found in all isolates of the ABU and UTI groups, respectively (Table 2). In addition, there was a high occurrence of the iron uptake related *fyuA* (ABU: 85%; UTI: 67%) and the biofilm related *agn43* (ABU: 85%; UTI: 67%); however, there was no significant difference between the ABU and UTI group with respect to the number of virulence factors (mean: ABU 3.3; UTI 2.5; *p* > 0.05). ExPEC was defined as *E. coli* isolates containing two or more of the following VAGs: P-fimbriae (*papA*/*papC*), S/F1C fimbriae (*sfa*/*foc*), Dr binding adhesins (*afa*/*dra*), capsule synthesis (*kpsMTII*) and aerobactin receptor (*iutA*) [27]. However, three ExPEC isolates were identified in each group.

### 3.4. Genetic Relationship

The phylogenetic tree showing SNP diversity across the collection of samples from this study illustrates that ABU and UPEC isolates of elderly did overall not occur in separate clusters in a SNP phylogeny (Figure 2). One cluster of two ABU isolates belonging to ST714 did not contain any UTI isolates, but ST714 has previously been identified in symptomatic UTIs [25]. The phylogenetic tree (Figure 2) illustrates closely related isolates in CC69: three isolates share a resistance phenotype, virulence profile, MLST (ST69) and phylogroup (D) and differ by only 5 and 32 SNPs, respectively. These isolates were all from three elderly people all living in the same nursing home. The short SNP distances indicate a common source or transmission between the study participants. We further compared whether this collection of isolates from symptomatic UTIs in the elderly represents different MLST types than a collection from younger women [26]. This revealed a large overlap in ST types between the two groups, but this should be confirmed in a larger study with more isolates.

### 3.5. Concentrations of Complement Component C3 in Urine Samples

Complement C3 and creatinine were determined in all urine samples with >10^5^ CFU/mL from patients with ABU (N = 19) and UTI (N = 16) and from 22 controls without bacteria (Figure 3A). The C3/Cr ratio was significantly increased in the ABU group (*p* = 0.005) and in the UTI group (*p* < 0.0001) as compared to the controls. However, there was no significant difference between the ABU and UTI groups (*p* > 0.05). In addition, we compared the C3/Cr ratio with the level of leukocyturia from dipstick measurements, which showed a clear correlation between increased C3/Cr and increased leukocyturia (R^2^ = 0.98, *p* = 0.001) (Figure 3B).

## 4. Discussion

Previous studies have reported that the prevalence of bacteriuria in elderly institutionalized individuals without indwelling catheters varies from 15 to 40% for men and 25 to 50% for women [28]. This concurs with our study, where we found a prevalence of 18.9% (N = 14/74) in men and women residing in nursing homes. In addition, we collected 140 urine samples from an activity center; a facility where primarily retired elderly individuals participated in various activities, such as card games, training and creative workshops. From this facility, we found a prevalence of ABU of only 4% (N = 5/140). By comparison, early studies of ABU prevalence in community populations have documented a consistent increase in cases with respect to age, with more than 5% in elderly men and 10% in elderly women [29].

During the first sample collection round from nursing homes, seven participants with ABU were identified, of which five provided additional urine samples three months later. Only two of these participants were colonized by the same bacterial species, whereas two were colonized by new species. Previous studies have shown that approximately one third of women with ABU converted from positive to negative cultures between surveys conducted at six-month intervals [30], and that colonization by *E. coli* is less likely to clear spontaneously than infections with other bacterial species [31,32].

The present dataset contains a limited number of isolates only from nursing homes—13 isolates of UTI/ABU and 3/13 were closely related. These data illustrate that outbreaks of pathogenic bacteria in care facilities and among the elderly for whom good personal hygiene may sometimes be difficult to orchestrate are likely. This illustrates the importance of strict infection control practices at such institutions. The genetic data combined show that ABU and symptomatic UTI in both the elderly and younger people cannot be predicted based on phylogroup, MLST or SNP phylogenies in the present dataset. This is in contrast to a study by Amarsy et al. [12], but was similar to the findings of other studies of *E. coli* from ABU and UTI [13,14,15]. In addition, we identified no significant differences between ABU and UTI regarding virulence-associated genes. The common virulence genes for urinary pathogenicity usually found in *E. coli* from UTI cases, i.e., *fimH*, *iutA* and *agn43*, were as common or even more common in the ABU isolates, which concurs with the previous studies [13,14,15].

The ABU *E. coli* isolates were significantly more resistant towards the antibiotics tested than the UTI isolates, and in addition, two of the isolates from the ABU group were ESBL-producers. Previous studies have shown that nursing home residents, more often than other patients, are colonized by resistant bacteria, such as methicillin resistant *Staphylococcus aureus* (MRSA), vancomycin resistant enterococci (VRE) [33] or multidrug resistant *E. coli* [34]. The acquisition of organisms of increased antimicrobial resistance in long-term care facilities is due to the increased frequency of antimicrobial treatment, the use of invasive devises (catheters and feeding tubes) and the fact that the residents are living in close proximity and participate in social activities [28,32]. Indeed, the majority of our ABU samples were collected from nursing homes, whereas UTI isolates were primarily from the GP. However, we did not identify an association between number of resistances and the institutions, as an equal number of resistance markers was carried in subjects from each institution (data not shown). However, we stress the relatively small dataset presented.

The level of complement component C3 relative to creatinine was significantly increased in the ABU and UTI groups compared to controls. There was no significant difference between the ABU and UTI groups, and the urinary C3 level could not be used to differentiate between the two groups. Despite the lack of symptoms in patients with ABU, the increased level of C3 indicates an increased activity in the host immune response in this patient group, corresponding to the correlation we found with respect to increased leukocyturia. In addition, recently Sundvall et al. [35] reported increased interleukin-6 (IL-6) levels in ABU samples compared to individuals without bacteriuria, supporting a conclusion of immune activation during ABU. Increased C3 has previously been shown to facilitate opsonization of *E. coli* and assist in the invasion of bladder cells in an in vitro bladder cell model, allowing the bacteria to form intracellular bacterial communities protected from host defense mechanisms [36,37]. The ability to invade bladder cells might also be used by ABU isolates to escape host defense mechanisms, which is supported by the fact that all ABU isolates carried equal numbers of virulence-associated genes, including *fimH*, an important weapon in the invasion of epithelial bladder cells [32]. The comparable levels of C3 in ABU and UTI patients raises the question of whether ABU should be considered as an infection that should be treated with antibiotics. However, not all cases in either the ABU or the UTI group had increased levels of C3 in the urine. Future studies should stratify patients—both ABU and UTI—with regard to C3 urine levels before and after antibiotic treatment in order to evaluate whether this parameter can be used as a marker both for those who need treatment and for the effect of treatment.

A weakness of our study was the relatively low number of ABUs, leading to a low number of *E. coli* isolates for closer study and for comparisons with UTI strains; therefore, we cannot conclude on the virulence or antibiotic resistance data between ABU and UTI. We experienced difficulties obtaining valid urine samples in nursing homes, where ABU and UTI were prevalent. This was easier at the activity center, but there the prevalence of ABU was low. The strength of our study is the strict definition of ABU based on two urine samples for each individual, and the sufficient number of urine samples to show the importance of complement C3 in urine, a novelty not previously reported for ABU.

## 5. Conclusions

In conclusion, the prevalence of ABU in community residing elderly ≥60 years old was low (4%), whereas a high prevalence was found in nursing home residents (18.9%); both prevalences were in concordance with published data. Due to the high prevalence of ABU and diagnostic challenges in differentiating ABU from UTI, a new diagnostic tool is needed. We found an increase in the level of human complement component C3 in individuals with ABU and UTI compared to healthy controls, and a significant correlation between urinary C3 levels and leukocyturia levels. While complement C3 cannot be used to differentiate ABU from UTI, the finding of similar levels of complement C3 in ABU as in UTI, along with a similar prevalence of *E. coli* virulence factors in the two groups, indicate that ABU may at least in some cases present a state of infection which needs treatment. This should be further investigated.

## Figures and Tables

**Figure 1 microorganisms-09-00390-f001:**
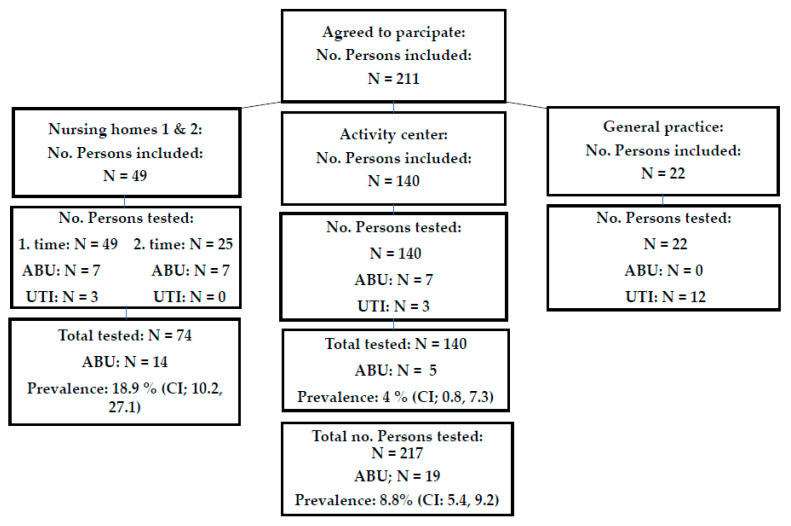
Diagram of sampling scheme and results of urine culture. ABU, asymptomatic bacteriuria; UTI, urinary tract infection. CI, 95% confidence interval.

**Figure 2 microorganisms-09-00390-f002:**
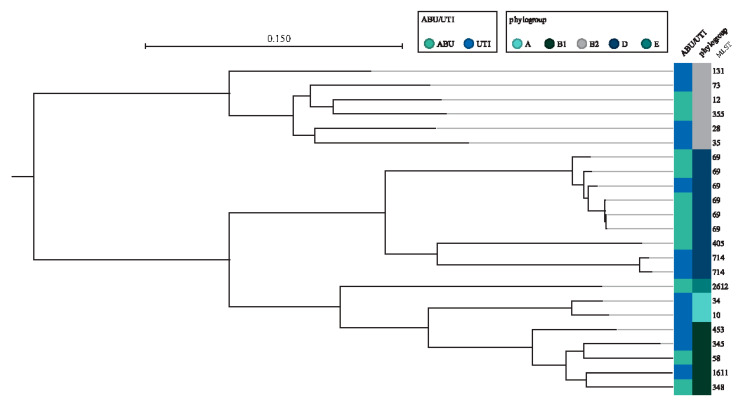
SNP phylogeny of *E. coli* isolates from asymptomatic bacteriuria (ABU) and urinary tract infections (UTI) based on whole-genome sequencing.

**Figure 3 microorganisms-09-00390-f003:**
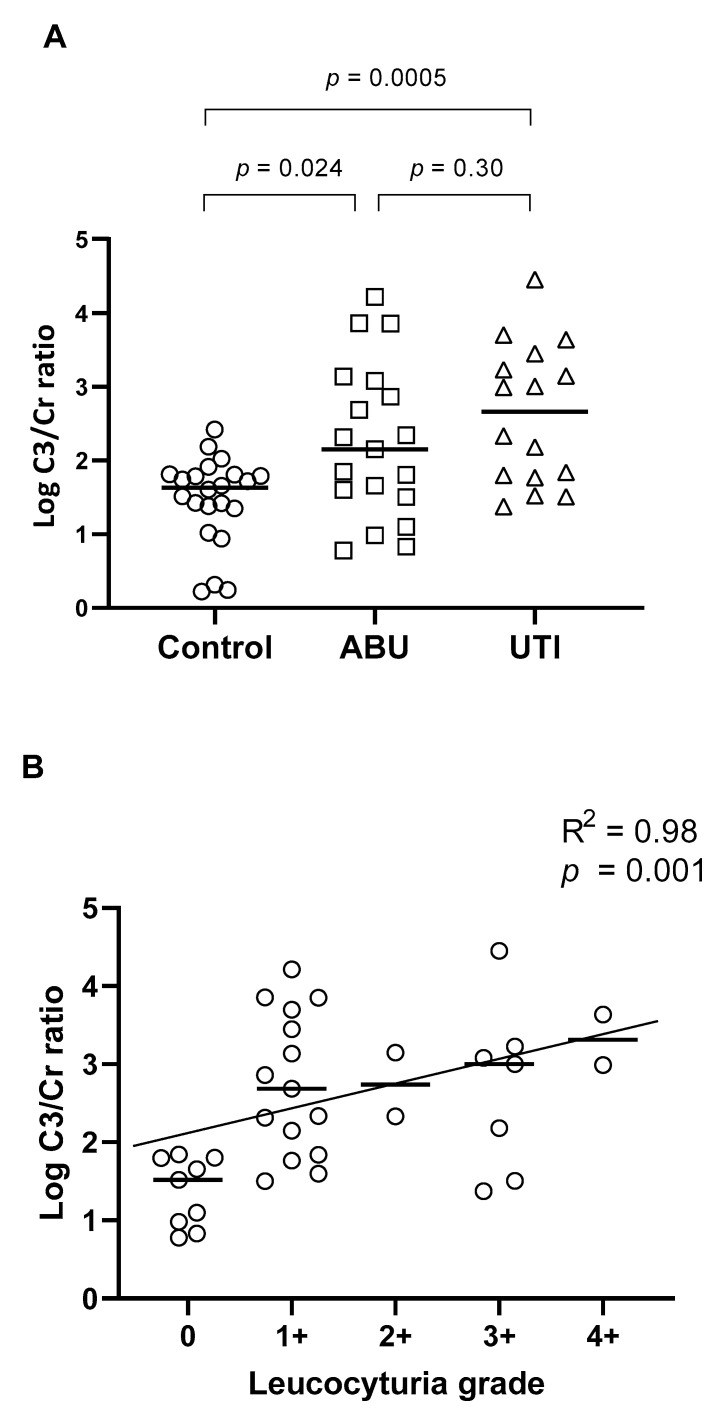
(**A**) Log complement C3/Cr (creatinine) ratio for urine from control (patients without symptoms and sterile urine), asymptomatic bacteriuria (ABU) and urinary tract infection (UTI). *p*-values for comparisons as indicated shown on top. (**B**) Linear correlation between leukocyturia measured as grades (0, no leucocytes, 4+, highest concentration of leucocytes; bars show medians) versus log C3/Cr ratio. Correlation coefficient squared (R^2^) and *p*-value indicated in figure.

**Table 1 microorganisms-09-00390-t001:** Prevalence of antibiotic resistance in *E. coli* isolates from patients with asymptomatic bacteriuria (ABU) and urinary tract infection (UTI). Numbers indicate number of *E. coli* isolates in total or number of isolates resistant towards antibiotic shown (percentage).

	Number of Isolates	Ampicillin	Cefuroxime	Ertapenem	Ciprofloxacin	Gentamicin	Tetracycline	Trimethoprim-Sulphamethoxa-zole
ABU	12 (100)	7 (58)	3 (25)	1 (8)	0 (0)	2 (17)	6 (50)	5 (42)
UTI	12 (100)	3 (25)	0 (0)	0 (0)	1 (8)	0 (0)	1 (8)	3 (25)
Total	24 (100)	10 (42)	3 (13)	1 (4)	1 (4)	2 (8)	7 (29)	8 (33)

**Table 2 microorganisms-09-00390-t002:** Distribution of the most common virulence genes identified in *E. coli* from urine in patients with asymptomatic bacteriuria (ABU) and urinary tract infection (UTI). For explanation of virulence genes, see text.

	N *	*fimH*	*traT*	*fyuA*	*kpsMTII*	*kpsK2*	*agnK12*	*Agn43*	*iutA*
ABU	12 * (100)	12 (100)	7 (58)	10 (83)	3 (25)	4 (33)	8 (67)	10 (83)	4 (33)
UTI	12 (100)	12 (100)	3 (25)	8 (67)	5 (42)	5 (42)	6 (50)	8 (67)	4 (33)
Total	24 (100)	24 (100)	10 (42)	18 (75)	8 (33)	9 (38)	14 (58)	18 (75)	8 (33)

* Number of isolates (percent).

## Data Availability

The supporting data of this study are located in figshare (https://figshare.com/articles/dataset/C3_content_in_urine_and_virulence_factors_in_E_coli/13898369 (accessed on 13 February 2021)). Raw reads from the sequencing are deposited in SRA under accession number PRJNA701152.

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
