# Peer review of "Asymptomatic Bacteriuria (ABU) in Elderly: Prevalence, Virulence, Phylogeny, Antibiotic Resistance and Complement C3 in Urine"

_microorganisms, 2021, doi:10.3390/microorganisms9020390_

Round 1

Reviewer 1 Report

Fleron Leihof et al, investigated the prevalence of ABU in elderly with focusing on the prevalence, virulence, phylogeny, antibiotic resistance and complement C3 in urine. The aim of the study is to assess the characteristics of E. coli isolates between ABU and UTI cases, as well as possible correlation between urinary levels of complement component C3 and ABU. The authors found an increase in the level of human complement component C3 in individuals with ABU and UTI compared to healthy controls, as well as a significant correlation between urinary C3 levels and leukocyturia levels.

Although the authors tried to investigate and assess all the above mentioned points, there are plenty of mistyping’s and scientific mistakes in this current format of manuscript.

Manuscript is not properly prepared and it seems it is still in a draft format and the research is not designed and performed correctly. The abstract is irregularly structured and data were not scientifically presented.

Major comments:

- Although authors performed some genotyping analyses to characterize the E. coli isolates, some major phenotypic traits (i.e., biofilm formation, motility, hemolysins, and proteases) of isolates are missing. I think the interpretation of the findings require clearer interpretation and analyses.

- The English in the present manuscript is not of publication quality and require major improvement. Therefore, a diligent editing is in order to fix the English language is necessary.

- References amended in the text do not follow the number of final references list!!

Minor comments:

- Abbreviations should be defined in parentheses the first time they appear in the abstract.

- Bacterial names must be written in their correct and precise format.

Author Response

Reviewer no. 1:

Comments and Suggestions for Authors

Fleron Leihof et al, investigated the prevalence of ABU in elderly with focusing on the prevalence, virulence, phylogeny, antibiotic resistance and complement C3 in urine. The aim of the study is to assess the characteristics of E. coli isolates between ABU and UTI cases, as well as possible correlation between urinary levels of complement component C3 and ABU. The authors found an increase in the level of human complement component C3 in individuals with ABU and UTI compared to healthy controls, as well as a significant correlation between urinary C3 levels and leukocyturia levels.

Although the authors tried to investigate and assess all the above mentioned points, there are plenty of mistyping’s and scientific mistakes in this current format of manuscript.

Authors response:

We have done our best  to correct mistypings and scientific mistakes

Manuscript is not properly prepared and it seems it is still in a draft format and the research is not designed and performed correctly. The abstract is irregularly structured and data were not scientifically presented.

Author response:

The abstract has been structured as suggested, and data have been corrected according to the results.

Major comments:

- Although authors performed some genotyping analyses to characterize the E. coli isolates, some major phenotypic traits (i.e., biofilm formation, motility, hemolysins, and proteases) of isolates are missing. I think the interpretation of the findings require clearer interpretation and analyses.

Author response:

The present study aimed to compare the E. coli genomicaly. Since there were no differences between genotypic or genetic traits of the E. coli isolates from the different patient categories, we did not pursue their phenotypic traits in this study. Instead, we find it more interesting in view of the results of the complement C3 study, to pursue the possible relationship between C3-binding to the E. coli cell wall and competition between increased phagocytosis by epithelial cells or neutrophils in the urine in a future study.

- The English in the present manuscript is not of publication quality and require major improvement. Therefore, a diligent editing is in order to fix the English language is necessary.

Author response:

The manuscript has been corrected for English language throughout.

- References amended in the text do not follow the number of final references list!!

Author response:

We completely agree that references did not follow the reference list; the manuscript amended to the reviewers was in complete dis-arrangement as compared to the submitted manuscript. A new reference (23) was added and the reference list and text-connected numbering has subsequently been corrected. The references already followed the instruction for authors.

Minor comments:

- Abbreviations should be defined in parentheses the first time they appear in the abstract.

Author response:

Corrected.

- Bacterial names must be written in their correct and precise format.

Author response:

Corrected.

Reviewer 2 Report

The publication raises an important issue regarding the decision to administer drugs in people with asymptomatic bacteriuria. The problem raised by the authors is ambiguous, as it concerns elderly people living in nursing homes, and thus with limited possibilities to signal their ailments. Additionally, they have unclear state of the immune system. The results of the study confirmed the it is necessary to carefully observe the patient's condition and  to repeat frequent urine tests in this group of people.

Author Response

Reviewer no. 2

Comments and Suggestions for Authors

The publication raises an important issue regarding the decision to administer drugs in people with asymptomatic bacteriuria. The problem raised by the authors is ambiguous, as it concerns elderly people living in nursing homes, and thus with limited possibilities to signal their ailments. Additionally, they have unclear state of the immune system. The results of the study confirmed the it is necessary to carefully observe the patient's condition and  to repeat frequent urine tests in this group of people.

Author response:

We thank the reviewer for the positive comments!

No further action taken.

Reviewer 3 Report

This is an interesting manuscript tackling the issue of symptomatic bacteriuria in the elderly; however, certain issues should be addressed. First and foremost, thorough proofreading is needed to amend all smaller mistakes and language issues throughout the paper. The Abstract is ill-formatted and the abbreviations should be avoided. On the other hand, in the text, all abbreviations should be initially stated in full (this is valid for PCR, IDSA and SNP, to name few examples).

From the epidemiological standpoint, the manuscript is valuable. However, the biggest problem in the manuscript is the conclusion that complement C3 response may be used to delineate infection and just bacteriuria, as the evidence is not compelling, i.e. the burden of evidence is not met. Such conclusions should be definitely toned down in this manuscript, especially in the Abstract section. The authors themselves note that not all cases in either the ABU or the UTI group had increased levels of C3 in the urine.

In the 'Materials and Methods' section, it is stated that the questionnaire also asked for age; however, only gender differences are shown in the results section in accordance with the presence of ABU and UTI. It is not clear whether the age stated in the 'Whole-genome sequencing' part of the Methods section is from the study sample (when the comparison with UTI isolates in Zealand, Denmark is mentioned), but it should also be emphasized and delineated in the 'Results' section of the manuscript.

It is also stated that in case of discrepancies between E. coli and Shigella dysenteriae, API-20 E was used after MALDI-TOF; however, it is not clear how many of these case there has been. Furthermore, why both EUCAST and CLSI breakpoints were used? If there are some differences that may influence the obtained results, this should be stated. Moreover, there are no information whether an informed consent was obtained from study participants. It should be clearly stated whether participants who were unable to give the confirmative sample were included in the analysis of the respective sampling round or not.

The 'Discussion' section should also address anbtibiotic stewardship issues that stem from resistance patterns that were found in the study. Figure 1 (sampling scheme and results) is substandard for a scientific paper, and should be make more legible (e.g., the 'plus-minus' sign should be used instead of an underlined plus sign). All references should be formatted in accordance with the journal rules. Bacterial names should always be italicized in references as well (e.g. this is not the case for reference 25 and 28), and there are formatting problems with certain names of the authors.

Author Response

Reviewer no. 3

Comments and Suggestions for Authors

This is an interesting manuscript tackling the issue of symptomatic bacteriuria in the elderly; however, certain issues should be addressed. First and foremost, thorough proofreading is needed to amend all smaller mistakes and language issues throughout the paper.

Author response:

Similar response as to reviewer no. 1. The manuscript has run through a detailed proofreading and mistakes and language issues corrected.

The Abstract is ill-formatted and the abbreviations should be avoided. On the other hand, in the text, all abbreviations should be initially stated in full (this is valid for PCR, IDSA and SNP, to name few examples)

Author response:

Abstract has been formatted, and abbreviations stated in full where needed.

From the epidemiological standpoint, the manuscript is valuable. However, the biggest problem in the manuscript is the conclusion that complement C3 response may be used to delineate infection and just bacteriuria, as the evidence is not compelling, i.e. the burden of evidence is not met. Such conclusions should be definitely toned down in this manuscript, especially in the Abstract section. The authors themselves note that not all cases in either the ABU or the UTI group had increased levels of C3 in the urine.

Author response:

We agree with these comments and the issue about complement C3 has been downplayed and corrected, both in the abstract and the discussion section.

In the 'Materials and Methods' section, it is stated that the questionnaire also asked for age; however, only gender differences are shown in the results section in accordance with the presence of ABU and UTI. It is not clear whether the age stated in the 'Whole-genome sequencing' part of the Methods section is from the study sample (when the comparison with UTI isolates in Zealand, Denmark is mentioned), but it should also be emphasized and delineated in the 'Results' section of the manuscript.

Author response:

The age issue has been mentioned in the results section, and Figure 2 and its legend has been corrected/re-structured (UTI and not UVI or EPEC-elderly)

It is also stated that in case of discrepancies between E. coli and Shigella dysenteriae, API-20 E was used after MALDI-TOF; however, it is not clear how many of these case there has been.

Author response:

Good point. Since there were no E. coli strains, that caused diagnostic problems, the API-20E part has been deleted.

Furthermore, why both EUCAST and CLSI breakpoints were used? If there are some differences that may influence the obtained results, this should be stated.

Author response:

We followed EUCAST breakpoints and have deleted CLSI and its reference.

Moreover, there are no information whether an informed consent was obtained from study participants. It should be clearly stated whether participants who were unable to give the confirmative sample were included in the analysis of the respective sampling round or not.

Author response:

The permission obtained from the ethics committee always includes informed consent; anyway, we have added a sentence about this issue. Regarding participants´ ability to provide samples, this was covered by the exclusion criteria mentioned in the M & M section.

The 'Discussion' section should also address anbtibiotic stewardship issues that stem from resistance patterns that were found in the study.

Author response:

We feel that we have already in the discussion mentioned the problems with antibiotic treatment or over-treatment.

Figure 1 (sampling scheme and results) is substandard for a scientific paper, and should be make more legible (e.g., the 'plus-minus' sign should be used instead of an underlined plus sign).

Author response:

Figure 1 has been corrected and reformatted. Confidence intervals are now shown as lowest and highest percentages.

All references should be formatted in accordance with the journal rules. Bacterial names should always be italicized in references as well (e.g. this is not the case for reference 25 and 28), and there are formatting problems with certain names of the authors.

Author response:

We did follow the journal rules as outlined in the author guideline. The reference list has been corrected, as mentioned above. All bacterial names have been italicized

Round 2

Reviewer 1 Report

I'm almost satisfied with the authors corrections, but I would suggest the authors to cite a number of recent and specific papers in the following sections:

In discussion section:

line 317 that defines fimH, an important weapon in the invasion of epithelial bladder cells.

Suggested refs:

- Sarshar, M.; Behzadi, P.; Ambrosi, C.; Zagaglia, C.; Palamara, A.T.; Scribano, D. FimH and Anti-Adhesive Therapeutics: A Disarming Strategy Against Uropathogens. Antibiotics 20209, 397.

- Scribano, D.; Sarshar, M.; Prezioso, C.; Lucarelli, M.; Angeloni, A.; Zagaglia, C.; Palamara, A.T.; Ambrosi, C. d-Mannose Treatment neither Affects Uropathogenic Escherichia coli Properties nor Induces Stable FimH Modifications. Molecules 202025, 316.

- Terlizzi ME, Gribaudo G, Maffei ME. UroPathogenic Escherichia coli (UPEC) Infections: Virulence Factors, Bladder Responses, Antibiotic, and Non-antibiotic Antimicrobial Strategies. Front Microbiol 2017, 15, 8:1566.

- Flores-Mireles, A., Walker, J., Caparon, M. et al. Urinary tract infections: epidemiology, mechanisms of infection and treatment options. Nat Rev Microbiol 2015, 13, 269–284.

In Material and Methods and Results sections:

Virulence genotyping of E. coli.

and

High prevalence of fimh and fyuA genes in E. coli isolates.

Suggested refs:

Ambrosi, C.; Sarshar, M.; Aprea, M.R.; Pompilio, A.; Di Bonaventura, G.; Strati, F.; Pronio, A.; Nicoletti, M.; Zagaglia, C.; Palamara, A.T.; et al. Colonic adenoma-associated Escherichia coli express specific phenotypes. Microbes Infect 2019, 21, 305–312.

Sarshar, M.; Scribano, D.; Marazzato, M.; Ambrosi, C.; Aprea, M.R.; Aleandri, M.; Pronio, A.; Longhi, C.; Nicoletti, M.; Zagaglia, C.; et al. Genetic diversity, phylogroup distribution and virulence gene profile of pks positive Escherichia coli colonizing human intestinal polyps. Microb. Pathog. 2017, 112, 274–278.

Minor comments:

In line 196: E. coli was the primary causative bacterium in both the ABU and UTI group, representing 63% and 75%. These percentages should contain numbers.

Table 1 and 2 should be modified to represent all the numbers together with percentage as well as total count for each table.

Author Response

Author´s response to reviewers (only No. 1 – others were satisfied):

Reviewer no. 1

Comments and Suggestions for Authors

I'm almost satisfied with the authors corrections, but I would suggest the authors to cite a number of recent and specific papers in the following sections:

In discussion section:

line 317 that defines fimH, an important weapon in the invasion of epithelial bladder cells.

Suggested refs:

- Sarshar, M.; Behzadi, P.; Ambrosi, C.; Zagaglia, C.; Palamara, A.T.; Scribano, D. FimH and Anti-Adhesive Therapeutics: A Disarming Strategy Against Uropathogens. Antibiotics 20209, 397.

- Scribano, D.; Sarshar, M.; Prezioso, C.; Lucarelli, M.; Angeloni, A.; Zagaglia, C.; Palamara, A.T.; Ambrosi, C. d-Mannose Treatment neither Affects Uropathogenic Escherichia coli Properties nor Induces Stable FimH Modifications. Molecules 202025, 316.

- Terlizzi ME, Gribaudo G, Maffei ME. UroPathogenic Escherichia coli (UPEC) Infections: Virulence Factors, Bladder Responses, Antibiotic, and Non-antibiotic Antimicrobial Strategies. Front Microbiol 2017, 15, 8:1566.

- Flores-Mireles, A., Walker, J., Caparon, M. et al. Urinary tract infections: epidemiology, mechanisms of infection and treatment options. Nat Rev Microbiol 2015, 13, 269–284.

In Material and Methods and Results sections:

Virulence genotyping of E. coli.

and

High prevalence of fimh and fyuA genes in E. coli isolates.

Suggested refs:

Ambrosi, C.; Sarshar, M.; Aprea, M.R.; Pompilio, A.; Di Bonaventura, G.; Strati, F.; Pronio, A.; Nicoletti, M.; Zagaglia, C.; Palamara, A.T.; et al. Colonic adenoma-associated Escherichia coli express specific phenotypes. Microbes Infect 2019, 21, 305–312.

Sarshar, M.; Scribano, D.; Marazzato, M.; Ambrosi, C.; Aprea, M.R.; Aleandri, M.; Pronio, A.; Longhi, C.; Nicoletti, M.; Zagaglia, C.; et al. Genetic diversity, phylogroup distribution and virulence gene profile of pks positive Escherichia coli colonizing human intestinal polyps. Microb. Pathog2017, 112, 274–278.

Authors response:

We found it most logical to include a new sentence: “In particular fimH and iutA are common virulence genes in E. coli found in the gut or in the urinary tract and important for adherence and penetration intracellularly [20,21]” under Material and Methods, line 130, where we included referral to two of the suggested references (Terlizzi et al. and Ambrosi et al.), which became references no. 20 and 21, and have renumbered succeeding references and revised in the text accordingly.

Minor comments:

In line 196: E. coli was the primary causative bacterium in both the ABU and UTI group, representing 63% and 75%. These percentages should contain numbers.

Authors response:

Numbers have been added together with percentages.

Table 1 and 2 should be modified to represent all the numbers together with percentage as well as total count for each table.

Authors response:

Table 1 and 2 have been revised accordingly, now including totals and percentages.

Reviewer 3 Report

The authors answered all queries and amended the manuscript where applicable.

Author Response

None - reviewer was satisfied